# The nuclear pore complex prevents sister chromatid recombination during replicative senescence

Paula Aguilera[1], Jenna Whalen[2,3], Christopher Minguet[1,3], Dmitri Churikov[1], Catherine Freudenreich [2], Marie-Noëlle Simon[1]* & Vincent Géli[1]*

The Nuclear Pore Complex (NPC) has emerged as an important hub for processing various types of DNA damage. Here, we uncover that fusing a DNA binding domain to the NPC basket protein Nup1 reduces telomere relocalization to nuclear pores early after telomerase inactivation. This Nup1 modification also impairs the relocalization to the NPC of expanded CAG/CTG triplet repeats. Strikingly, telomerase negative cells bypass senescence when expressing this Nup1 modification by maintaining a minimal telomere length compatible with proliferation through rampant unequal exchanges between sister chromatids. We further report that a Nup1 mutant lacking 36 C-terminal residues recapitulates the phenotypes of the Nup1-LexA fusion indicating a direct role of Nup1 in the relocation of stalled forks to NPCs and restriction of error-prone recombination between repeated sequences. Our results reveal a new mode of telomere maintenance that could shed light on how 20% of cancer cells are maintained without telomerase or ALT.

[1] Marseille Cancer Research Center (CRCM), U1068 Inserm, UMR7258 CNRS, Aix Marseille University, Institut Paoli-Calmettes Equipe labellisée Ligue, 27 bd Lei Roure, Marseille, France. [2] Department of Biology, Tufts University, 200 Boston Ave, Medford, MA 02155, USA. [3] These authors contributed equally: Jenna Whalen, Christopher Minguet. *email: marie-noelle.simon@inserm.fr; vincent.geli@inserm.fr

Telomeres are nucleoprotein structures that protect the ends of linear eukaryotic chromosomes against degradation, end-to-end fusions and homologous recombination (HR). They consist of G-rich repetitive DNA sequences with a terminal 3′ single-strand overhang[1]. Capping of telomeres is necessary to avoid their recognition as double-strand breaks (DSBs) and resulting genome instability[2]. In *Saccharomyces cerevisiae*, telomeres consist of an array of about 300-bp of $TG_{1-3}$ repeats and a 12–14-nucleotide-long 3′ single-strand overhang. The Rap1 protein wraps the double-stranded telomeric DNA to inhibit end fusion[3] and recruits Rif1 and Rif2 that limit HR[4–6]. The 3′ overhang is bound by Cdc13, a subunit of the CST (Cdc13/Stn1/Ten1) complex[7]. The specific sequence and chromatin properties of telomeres create a challenge for the DNA replication machinery that requires accessory factors at telomeres in both yeast and mammals[8,9].

Telomere length is carefully regulated, and is maintained by the reverse transcriptase telomerase to bypass the end replication problem[10]. As in metazoans, inactivation of telomerase in yeast leads to progressive shortening of telomeres until they reach a critical length that impairs their capping function. When telomeres become critically short, unprotected telomeres elicit a DNA damage response (DDR), recruit Mec1[ATR] and activate a permanent G2/M arrest[11,12] leading to replicative senescence. However, telomerase is not only required to replenish chromosome ends upon their gradual erosion but also essential to counteract replication-induced damage at telomeres[13–15]. Therefore, in the absence of telomerase, telomere replication becomes imminently dependent on HR factors. As a consequence, the deletion of *RAD52* or *RAD51* dramatically accelerates senescence[16], although how HR resolves replication stress remains poorly understood.

The progressive loss of the growth capacity observed upon telomerase inactivation thus stems from two distinct types of damage, i.e., stochastic replication fork stallings that induce transient cell-cycle arrests soon after telomerase inactivation[17,18] and gradual erosion of the telomeres from their termini that eventually induces permanent cell-cycle arrests[12]. A small fraction of the permanently arrested cells can bypass senescence and resume divisions by regenerating functional telomeres through recombination[16]. Two types of survivors are described in budding yeast. Type I survivors amplify Y′ subtelomeric sequences separated by short interstitial $TG_{1-3}$ repeats while maintaining very short telomere ends. Type II survivors have very long $TG_{1-3}$ tracts, heterogeneous in length[16] that resemble telomeres in the cancer cells that maintain their telomeres via alternative lengthening of telomeres (ALT). Both types of repair use break-induced replication (BIR) dependent on Rad52, Pol32 and Pif1[16,19,20]. Type I recombination requires in addition Rad51, Rad54 and Rad57, while Type II recombination depends on Rad59, the MRX (Mre11/Rad50/Xrs2) complex, Sgs1 and Sae2[21–25].

DNA damage can be repaired by multiple and sometimes redundant mechanisms with variable capacities to preserve genetic information and genome stability. Repair of DNA damage appears to be spatially segregated within the nucleus[26,27]. In budding yeast, persistent DSBs, usually those that lack a homologous sequence for repair, are often shifted to the vicinity of the nuclear envelope (NE)[28,29]. One key structure at the NE is the nuclear pore complex (NPC) that forms an aqueous channel embedded in the nuclear membrane. NPCs consist of a central ring structure in the plane of the NE that extends as cytoplasmic filaments on one side and the nuclear basket on the other side[30]. Integrity of the NPC is required for proper response to DNA damage, as mutating NPC components confers sensitivity to DNA-damaging agents[31–33]. Similarly to persistent DSBs, highly eroded and poly-SUMOylated telomeres shift to the nuclear pores[34,35].

We showed that relocalization of eroded telomeres to the nuclear pores favors type II recombination[35]. This observation was in line with the view that NPC relocalization promotes non-canonical repair of congested DNA structures using error-prone last chance mechanisms such as BIR or microhomology-mediated end-joining[36]. Nevertheless, the role of the nuclear pore might extend to other types of telomere damage as some damaged telomeres are detected at the NPCs early after telomerase inactivation and might represent telomeres under replication stress[34]. Indeed, the replication forks that confront a barrier formed by expanded CAG/CTG triplet repeats transiently shift to the NPCs, a mechanism that prevents repeat fragility and instability and is thus hypothesized to promote fork restart[37].

In this study, we uncover that modification of the NPC basket protein Nup1 impairs the relocalization to NPCs of the telomeres under replication stress in telomerase-negative as well as expanded CAG/CTG triplet repeats in telomerase-positive cells. Strikingly, in telomerase-negative cells expressing modified Nup1, telomeres are maintained for generations at a short but functional length. This telomere maintenance mechanism that relies on non-conservative exchanges between sister chromatids is sufficient to avoid the proliferative decline and telomere shortening-driven crisis normally observed in telomerase-negative cells. Collectively, our results unveil an unsuspected role of the NPC in suppressing sister-chromatid recombination (SCR) at telomeres and reveal a new mechanism of telomere maintenance.

## Results

**Fusion of LexA to Nup1 prevents senescence in the absence of telomerase.** In order to address the role of the NPC in controlling telomere recombination during replicative senescence[35], we constructed a strain in which Nup1 was C-terminally tagged at its native locus with the DNA binding protein LexA (Fig. 1a). Our initial idea was to develop an assay to tether genomic regions tagged with LexA-binding sites to the NPC in telomerase-negative cells. In contrast to the deletion of *NUP1*, the Nup1-LexA fusion was viable and did not impair cell proliferation (Supplementary Fig. 1a, b). It also showed no additive effects with the absence of the mitotic exit regulator Bub2 (Supplementary Fig. 1b) that aggravates the growth defect of *NUP1*-deficient cells[38]. Importantly, the SUMO-protease Ulp1 was properly localized to the nuclear rim in *nup1-LexA* cells (Supplementary Fig. 1c). Furthermore, quantitative assessment of the nuclear export of poly (A)+-RNA and of the nucleocytoplasmic transport of proteins did not reveal any gross alterations in *nup1-LexA*-expressing cells as compared to WT cells (Supplementary Fig. 1d–f). Thus, in all these respects, the *nup1-LexA* strain did not have the defects of the *NUP1*-deficient strain.

The haploid *nup1-LexA* strain was crossed with a telomerase-negative *est2Δ* strain maintained alive with an *EST2*-expressing plasmid. Haploid *est2Δ* spore clones were isolated after micromanipulation of the spores on plates and then propagated in liquid cultures for about 120 population doublings (PDs) via serial dilutions every 24 h. The growth capacity of the clonal cell populations was assessed by measurement of the cell density every day. As expected, proliferation of the control *est2Δ* cells declined progressively until the culture entered telomere-erosion-driven crisis after about 70 PDs before formation of the survivors (Fig. 1b). In striking contrast, *est2Δ nup1-LexA* clones greatly attenuated the loss of proliferative capacity and essentially bypassed senescence (Fig. 1b). Importantly, these results were obtained in *est2Δ nup1-LexA* cells lacking any LexA-binding sites.

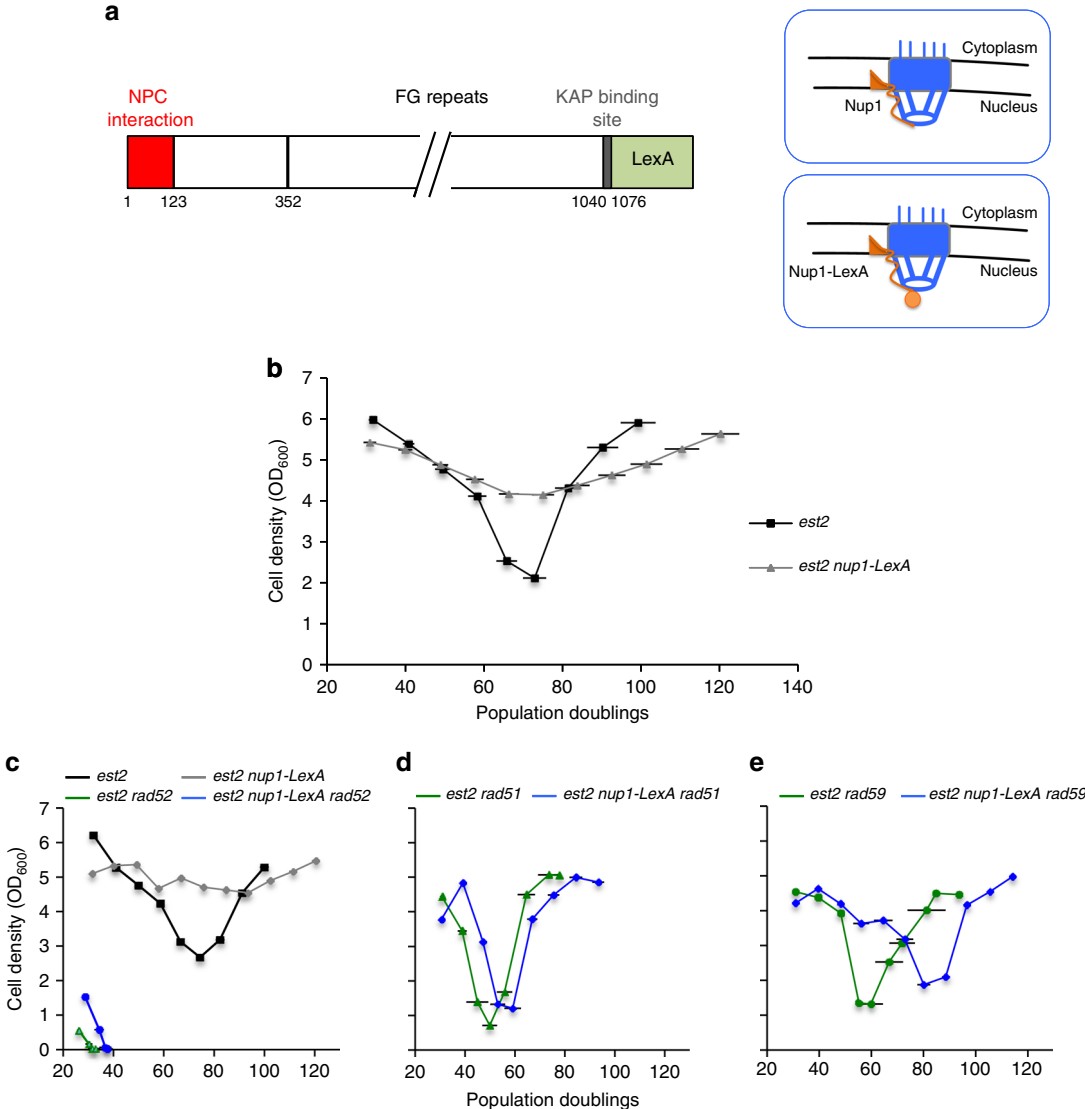

**Fig. 1 Attenuation of senescence in *est2Δ nup1-LexA* cells. a** Schematic structure of the nuclear basket protein Nup1 fused to LexA. **b** Mean senescence profiles of the *est2Δ* (*n* = 22) and *est2Δ nup1-LexA* (*n* = 28) clones analyzed in the course of this study. Each clone was isolated by sporulation of a heterozygous diploid strain. Each spore colony was then propagated in liquid culture through daily serial dilutions. OD$_{600}$ was measured every day to estimate the cell density reached in 24 h. PD numbers were estimated from the initial spores. **c** Mean senescence profiles of *est2Δ* and *est2Δ nup1-LexA* clones in the absence of *RAD52* (*n* = 4 and 8, respectively). Control *est2Δ* (*n* = 7) and *est2Δ nup1-LexA* (*n* = 5) curves are from the clones used in this specific experiment. **d** Mean senescence profiles of *est2Δ* and *est2Δ nup1-LexA* clones in the absence of *RAD51* (*n* = 4 and 7, respectively). **e** Mean senescence profiles of *est2Δ* and *est2Δ nup1-LexA* clones in the absence of *RAD59* (*n* = 4 and 6, respectively). Error bars indicate SD.

To determine whether the *nup1-LexA* allele confers a defect in DNA damage checkpoint activation, we measured the sensitivity of telomerase-positive *nup1-LexA* cells to hydroxyurea (HU), methyl methanesulfonate (MMS) and UV irradiation. We detected neither increased sensitivity nor resistance of *nup1-LexA* cells to HU, MMS or UV irradiation (Supplementary Fig. 1g). Accordingly, untreated *nup1-LexA* cells did not display auto-phosphorylation of Rad53 as observed in the presence of MMS (Supplementary Fig. 1h).

Taken together, these data indicate that although the Nup1-LexA fusion protein appeared largely functional with respect to Nup1 canonical functions, its expression suppressed the senescence phenotype of the telomerase-negative cells.

**Senescence bypass by the *est2Δ nup1-LexA* cells requires HR factors**. We sought to determine whether the ability of *nup1-LexA* cells to attenuate senescence requires HR components. We first

deleted *RAD52* in *est2Δ nup1-LexA* cells and re-examined their growth. *RAD52* deletion induced a premature and definitive growth arrest in *nup1-LexA est2Δ* cells as it did in the control cells (Fig. 1c). In contrast, *rad52Δ* did not affect growth of *nup1-LexA* cells expressing telomerase (Supplementary Fig. 1i). We next tested the role of Rad51 and Rad59 that define two distinct pathways of Rad52-dependent HR[39]. We observed that the role of HR in maintaining growth of *est2Δ nup1-LexA* cells relied on Rad51 as senescence was no longer attenuated by *nup1-LexA* in its absence (Fig. 1d). The rescue depended on Rad59 as well but to a lesser extent, as senescence was re-established but was delayed (Fig. 1e). Taken together, these results suggest that steady proliferation of the *est2Δ nup1-LexA* cells was dependent on HR.

**Nup1-LexA impairs the localization of the early telomere damage**. We showed that accumulation of single-strand DNA at a single telomere is sufficient to produce a fluorescent focus in cells

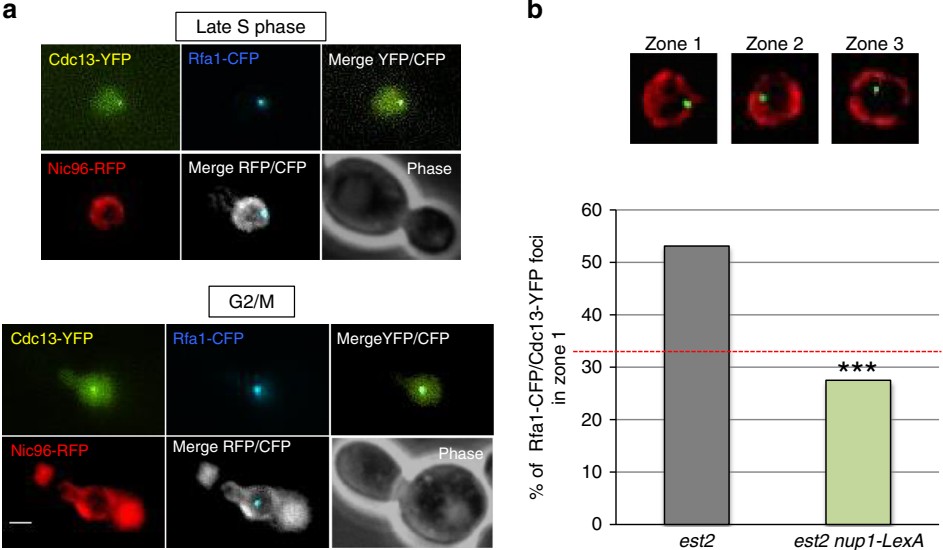

**Fig. 2 Nup1-LexA alters the localization of damaged telomeres during senescence. a** Damaged telomeres were detected as foci containing both Cdc13-YFP and Rfa1-CFP and localized relative to Nic96-RFP that marks the nuclear periphery. Representative images of late S (upper panel) and G2/M (lower panel) cells are shown. Scale bar: 2 μm. **b** The localization of the Cdc13-YFP/Rfa1-CFP foci in the three zones of equal area of the nucleus marked by Nic96-RFP (upper panel) was scored in late S and G2/M cells. Cells were imaged after one restreak from spore colonies (about 35 PDs from the initial spores). The percentages of zone 1 foci are from analysis of two independently isolated $est2\Delta$ ($n = 96$ cells) and $est2\Delta$ $nup1$-LexA ($n = 120$ cells) clones. The analysis of the repartition of Rfa1/Cdc13 foci between the three equal zones of the nucleus is shown for individual clones in Supplementary Fig. 2a. Statistical differences were determined by a Fisher's exact test (\*\*\*$p = 0.0001$).

expressing Cdc13-YFP[34]. Cdc13 foci are detected as soon as 20 PDs after telomerase inactivation and localize to the nuclear periphery. We inferred that these early Cdc13-YFP foci that are also enriched in Rfa1 may represent telomeric stalled replication forks[40]. To determine whether expression of Nup1-LexA interferes with the localization of early telomere damage, we performed a zoning assay in $est2\Delta$ and $est2\Delta$ $nup1$-LexA cells. In this assay, the nucleus is divided into three zones of equal volume, and the position of a given focus relative to the nuclear periphery, here defined by Nic96-RFP, is scored[41]. To visualize telomeric stalled forks, we analyzed the colocalization of Rfa1-CFP foci with Cdc13-YFP foci from two independently isolated $est2\Delta$ and $est2\Delta$ $nup1$-LexA clones after one restreak on yeast extract peptone dextrose (YPD) plates (about 30 PDs after telomerase inactivation) (Fig. 2a). The localization of the Cdc13-YFP/Rfa1-CFP foci was then scored in S/G2-M cells into one of three nuclear zones (Supplementary Fig. 2). Remarkably, the peripheral enrichment of the Cdc13-YFP/Rfa1-CFP foci observed in the $est2\Delta$ cells was totally abolished in the $nup1$-LexA-expressing cells (Fig. 2b and Supplementary Fig. 2).

**Nup1-LexA impairs triplet repeat stability and relocalization to NPC.** It was proposed that replication fork repair can occur either in the interior of the nucleus or at the nuclear periphery depending on the state of the arrested fork[42]. HU-stalled forks or forks that encounter a nick do not localize to the nuclear pore[29,43] while forks that encounter a naturally occurring barrier such as expanded CAG trinucleotide repeats do[37]. We therefore tested whether the localization of CAG sequences containing 130 repeats tagged with a $lacO$ array was altered in the presence of Nup1-LexA in telomerase-positive cells. To this purpose, we scored in S-phase the position of GFP-LacI foci relative to the nuclear periphery[37] (Fig. 3a). Samples were taken after each growth step to confirm that the tract length was not contracted in the cells used to start the assay (see Methods). Strikingly, the peripheral enrichment of the repeat locus observed in the WT strain was lost in the $nup1$-LexA-expressing cells (Fig. 3b)

suggesting that the fusion protein induced a general defect in the relocalization to the NPC of replication stress-induced damage. To further investigate the consequences of $nup1$-LexA expression in the processing of replication-induced damage, we monitored instability (expansions and contractions) (Fig. 3c) and fragility (breakage) (Supplementary Fig. 3a) of long CAG/CTG repeat tracts using established assays[37]. Expression of Nup1-LexA did not exacerbate the fragility of either CAG-70 or CAG-155 expanded repeats (Supplementary Fig. 3b). In contrast, it significantly affected the stability of the repeats and favored contractions at the expense of expansions (Fig. 3c). Notably the contractions were strongly reduced by inactivation of Rad52 (Fig. 3c), suggesting that $nup1$-LexA expression unleashes an HR pathway that is normally restricted when stalled forks at CAG repeats are relocalized to the NPC.

**Telomeres are maintained at a minimal length in $est2\Delta$ $nup1$-LexA cells.** To assess the state of telomeres during replicative senescence in telomerase-negative cells expressing Nup1-LexA, telomere length was analyzed by Southern blot at different time points of the senescence in several independently isolated clones. Figure 4 shows representative Southern blots of three of these clones. In all the clones analyzed, telomeres shortened as a function of the PDs in $est2\Delta$ $nup1$-LexA cells as they did in the control $est2\Delta$ cells. As expected, the liquid cultures of $est2\Delta$ cells after the senescence crisis were dominated by type II survivors because of their growth advantage over type I survivors (Fig. 4a, b). In striking contrast, $est2\Delta$ $nup1$-LexA clones maintained short tracts of $TG_{1-3}$ repeats for up to 130 PDs (Fig. 4b). Out of the 25 independent clones analyzed during the course of this study, six maintained short telomere tracts without any signs of either Y′ or $TG_{1-3}$ repeat amplification (Fig. 4b, clone #6). Nine showed some bands corresponding to type II recombination at the late time points while still maintaining short terminal $TG_{1-3}$ repeats at some telomeres (Fig. 4b, clone #A11). The remaining ten clones displayed Y′ amplification reminiscent of type I recombination at late time points (Fig. 4b, clone #A45). These results are consistent

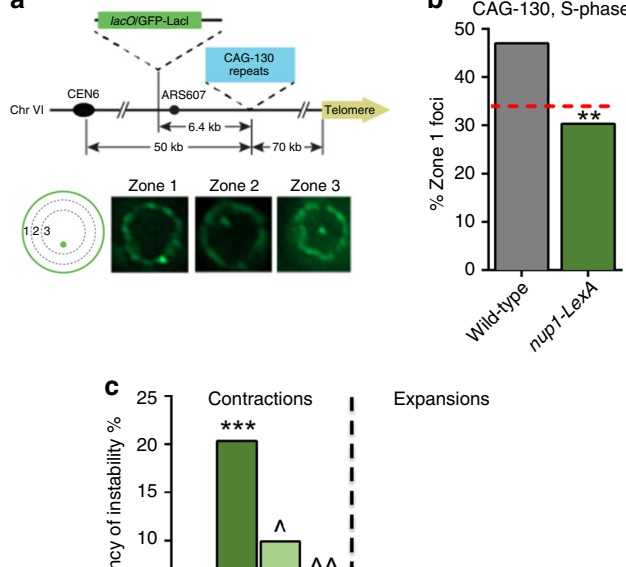

c

| | CAG-70 | | | | Total # of colonies tested |
|---|---|---|---|---|---|
| | Contractions | | Expansions | | |
| | # | % | # | % | |
| Wild-type | 14 | 5.0 | 9 | 3.2 | 279 |
| *nup1-LexA* | 24 | 20 | 1 | 0.85 | 118 |
| *nup1-LexA rad52Δ* | 13 | 9.9 | 3 | 2.3 | 131 |
| *rad52Δ* | 7 | 6.0 | 3 | 2.6 | 116 |

**Fig. 3 Nup1-LexA affects localization and stability of CAG trinucleotide repeats. a** Yeast chromosome VI containing 130 integrated CAG repeats and a *lacO* array that binds the LacI-GFP protein. S-phase cells were imaged, and the location of the foci was scored into one of three zones of equal area, using the GFP-Nup49 signal to mark the nuclear periphery. **b** Percent of zone 1 foci for CAG-130 S-phase cells (283 cells for wild-type, 142 cells for *nup1-LexA*). Statistical differences were determined by a Fisher's exact test (**$p = 0.0012$). **c** Frequency of contractions and expansions for CAG-70 repeats in wild-type and *nup1-LexA* cells. The frequency was determined by analysis of PCR amplicon length on a high-resolution fragment analyzer gel system, using PCR primers that flank the CAG-70 tract located on a YAC (see Fig. S6A); (***) $p = 0.0001$ compared with wild-type by Fisher's exact test; (^) $p = 0.0313$; (^^) $p = 0.0017$ compared with *nup1-LexA* by Fisher's exact test.

with a scenario where telomeres do not reach the critical length that induces permanent checkpoint activation but instead are maintained at a minimal length compatible with cell proliferation, and in so doing either prevent or postpone the types of recombination that lead to survivor formation.

**Telomeres are maintained by recombination with the sister chromatid.** To further investigate the type of repair occurring at telomeres in *est2Δ nup1-LexA* cells, we used telomere PCR to clone and sequence a subset of one telomere (*TelVI-R*) from clonal populations before cells enter the nadir of senescence using a specific *TELVI-R* probe that hybridizes just upstream the TG$_{1-3}$

repeats (Supplementary Fig. 4a, Day 5). Each sequence was compared with the reference sequence obtained from the same clone at the earliest time point (Day 1). The telomerase of *S. cerevisiae* has the peculiarity to add TG$_{1-3}$ repeats of variable length[44], and consequently the sequence of each individual telomere is unique and stable over generations provided that its distal region is not elongated by telomerase or modified through recombination with another telomere. Alignment of multiple copies of *TelVI-R* from *est2Δ* cells showed that, as expected, a large majority (94%) of the sequences matched perfectly with the reference sequence (Fig. 5a). In line with previous publication[22], 6% of the telomeres showed sequence divergence in the distal region that might stem from rare recombination events or from technical reasons related to PCR amplification and subcloning[45]. In contrast, more than a quarter of *TelVI-R* sequences from *est2Δ nup1-LexA* could not be perfectly aligned with the reference sequence over their total length (results from two independent clones are shown in Fig. 5a and Supplementary Fig. 4a–c).

Careful analysis of each misaligned part of the sequences showed that they can be all matched to the reference sequence of *TELVI-R* provided that either a gap or an insertion was introduced (Fig. 5b, clones a1 and e1, respectively). Because each telomere sequence is unique, this finding ruled out a possibility that seemingly divergent telomere sequences were a result of recombination with another telomere. Instead, gapped alignment is more consistent with SCR events. Notably, deletions always occurred within repeated motifs at the breakpoint (Fig. 5b, clone a1, repeated motif indicated by blue sequence). The occurrence of insertions was evident by starting the alignment upstream from the point of divergence (Fig. 5b, clone e1) indicating that duplication occurred during repair. Equal SCR normally generates repair products without change in DNA sequence. At telomeres, however, the nature of the repeats provides conserved motifs at several positions where a D-loop can be initiated (Fig. 5c). Shifted annealing with an upstream or downstream conserved motif would therefore produce a shorter or elongated telomere, respectively (Fig. 5c). In vivo, a 5- to 8-bp homologous tail at the 3′ end appears to be sufficient to assure recombination even in the presence of single-base-pair mismatches if the heteroduplex can be extended[46]. These sequencing results unveil a new mode of telomere maintenance (distinct from type I and II recombination), which operates early after telomerase inactivation in cells expressing *nup1-LexA* but is likely largely repressed in *est2Δ NUP1*+ cells.

Interestingly, although the helicase/ubiquitin ligase Rad5 acts in parallel to HR to delay senescence[47], *RAD5* deletion did not affect the continuous growth of *est2Δ nup1-LexA* cells (Supplementary Fig. 4d). This result suggests that Nup1-LexA overrides the requirement for the Rad5-dependent template switch branch of the DNA damage tolerance (DDT) pathway. In contrast, the helicase Srs2 played a key role in the continuous growth of *est2Δ nup1-LexA* cells, whereas its deletion only slightly affected the rate of senescence in *est2Δ* control cells (Supplementary Fig. 4e and ref. [48]). This result is in line with the role of Srs2 in promoting replication fork restart via template exchange at a natural protein/DNA replication barrier[49]. Srs2 helicase activity has also been shown to be important for fork restart at an expanded CAG/CTG tract fork barrier[50]. Finally, we found that Pif1 was required for the steady growth of *est2Δ nup1-LexA* cells suggesting that BIR is involved in maintaining functional telomeres in the absence of telomerase (Supplementary Fig. 4f). From these data, we conclude that Nup1-LexA expression confers permissive conditions for telomere repair by SCR that is normally restricted in WT cells.

These data therefore raised the possibility that Nup1-LexA favors repair of telomeres by SCR by hindering their

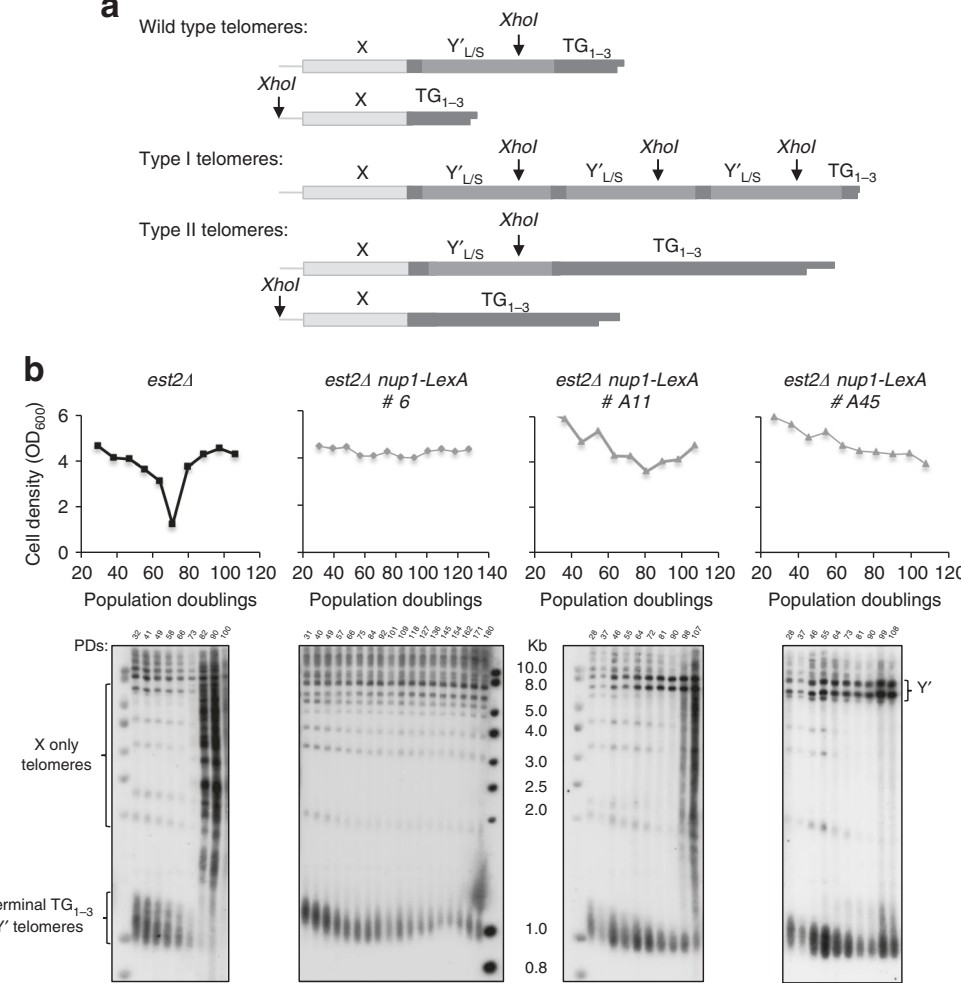

**Fig. 4 Telomere length is maintained at a minimal length in *nup1-LexA* cells. a** Schematic representation of wild-type, type I and type II telomeres. All telomeres contain one X element in the subtelomeric region. In addition, about two-third of the telomeres contain from one to four long (L) or short (S) subtelomeric sequences called Y′ separated by short interstitial $TG_{1-3}$ repeats, both being amplified in type I survivors. Positions of the *XhoI* sites are shown. **b** Senescence profiles of one *est2Δ* clone (black) and three independent representative *est2Δ nup1-LexA* clones (gray). Telomere length and survivor formation in the same replicative senescence experiment were monitored by $TG_{1-3}$ probed Southern blots of *XhoI*-digested DNA.

relocalization to the NPC. To further address this possibility, we forced the localization of one specific telomere to the NPC in cells expressing Nup1-LexA. To this end, we introduced eight LexA-binding sites proximal to telomere VI-R in *est2Δ nup1-LexA* cells (Fig. 6a and ref. [35]). The supplementary Fig. 5b shows the mean senescence curve of 11 independent *est2Δ nup1-LexA TelVI-R-8LexAbs* clones. As expected, the presence of a single telomere tethered to the NPC was not sufficient to significantly drive *est2Δ nup1-LexA* cells to senescence crisis. We thus evaluated the frequency of *TelVI-R* recombination events in *est2Δ nup1-LexA TelVI-R-8LexAbs* cells by sequencing a subset of telomeres as described above. Analysis of two independent clones showed that the occurrence of rearrangements was decreased from 26.5–28% to only 9–16% at *TelVI-R-8LexAbs* (Fig. 6b, c and Supplementary Fig. 5c). Southern blot analysis of terminal restriction fragments containing TelVI-R revealed that tethering TelVI-R to the pore increased the frequency of type II recombination events (Supplementary Fig. 5d, e). Together these results suggest that tethering a single telomere to the pore decreases SCR and the peculiar telomere maintenance mode observed in *nup1-LexA* cells. Moreover, it restores the odds of type II recombination typical to telomeres in *est2Δ* cells with unmodified Nup1.

**Rad53 is required to sustain growth of *est2Δ nup1-LexA* cells.** As mentioned above, stochastic transient cell cycle arrests occur in the heterogeneous population of senescing cells that are distinct from the terminal arrest induced by critically short telomeres[18]. Transient and terminal arrests are both dependent on checkpoint activation[18]. In particular, Rad53 was shown to have multiple roles at stalled replication forks including replisome stabilization and fork restart[51,52]. Rad53 was also proposed to release transcribed genes from the NPC when a replication fork encounters nuclear pore-gated transcripts[53]. We tested the role of the checkpoint effector kinase Rad53 in sustaining the growth of the *est2Δ nup1-LexA* cells by evaluating the senescence profiles of *est2Δ nup1-LexA* clones carrying the checkpoint deficient *rad53-K227A* allele. Strikingly, inactivation of Rad53 completely abolished the effect of Nup1-LexA on senescence and telomere maintenance (Supplementary Fig. 6a). This shows that the activity of Rad53 is required for SCR at telomeres in *est2Δ nup1-LexA* cells.

We also tested whether the SUMO pathway is involved in the unusual SCR occurring in the presence of Nup1-LexA. To this purpose, we introduced deletions of the two main E3 SUMO-ligase genes (*SIZ1* and *SIZ2*) in *est2Δ nup1-LexA* cells and monitored the senescence profiles of independent clones. Deleting

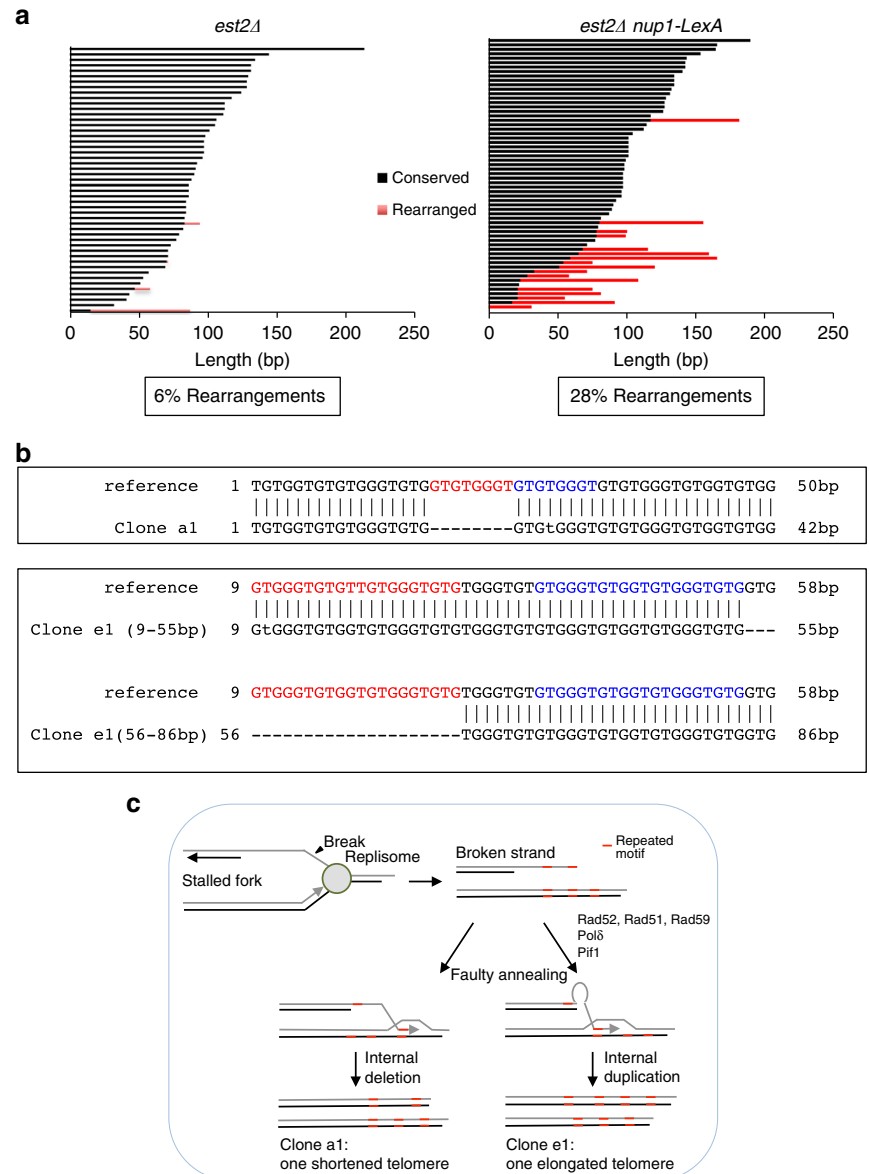

**Fig. 5 Analysis of telomeres by DNA sequencing. a** Telomeres VI-R were amplified and sequenced from clonal populations of *est2Δ* and *est2Δ nup1-LexA* cells using a specific primer after about 65 population doublings from the initial spores (Day 5) and align with the reference sequence obtained at Day 1 (see Supplementary Fig. 4a for the senescence profiles). Each bar represents an individual telomere. Bars are sorted by the length of the centromere-proximal sequence that is identical to the reference sequence (black). The red bars show the length of the distal rearranged sequence that cannot be continuously aligned with the reference. **b** Examples of rearrangements. Upper panel, deletion: *TelVI-R* from the clone a1 aligns with the reference provided that a 8-bp-long gap is introduced after the 17th nucleotide. Note the repeated motif (in red and blue) at and after the gap. Lower panel, insertion: The 55 first nucleotides of the clone e1 align perfectly with the reference. The last 60 nucleotides (56–86 are shown) cannot be aligned with the reference except if starting at position 29 of the reference instead of 56. This suggests that part of the sequence has been duplicated. Note the presence of the motif (in blue) at the end of the conserved sequence that is repeated just upstream the point of misalignment (in red). **c** Model of non-conservative HR-dependent repair between sister chromatids. The sequence of the telomeres contains motifs that are repeated several times. Thus a 3′ terminal end with such a motif can anneal at different positions in the sister chromatid to initiate repair leading to internal deletion (left part) or duplication (right part).

*SIZ1* and *SIZ2* did not alter the attenuated senescence profile conferred by the expression of Nup1-LexA (Supplementary Fig. 6b) indicating that SUMOylation is not a major actor of the repair pathway, that is at play in *nup1-LexA* cells.

**Truncation of the Nup1 C-terminal phenocopies the *nup1-LexA*.** Altered telomere recombination in the presence of Nup1-LexA might stem from either steric interference instigated by the C-terminal fusion or from the presence of an ectopic DNA binding domain at the pore. To distinguish between these two possibilities, we tested several alternative C-terminal Nup1 fusions. Neither a 13-myc (Supplementary Fig. 7a) nor a CFP tag (Supplementary Fig. 7b) attenuated replicative senescence. In contrast, fusion of the *lac* repressor LacI to the C-terminal end of Nup1 conferred a phenotype similar to *nup1-LexA* in the absence of telomerase, i.e., attenuated senescence (Supplementary Fig. 7c) and impaired survivor formation (Supplementary Fig. 7d).

The apparent dependency of the effect on the presence of a DNA binding moiety (LexA or LacI) initially suggested that attenuation of senescence required the DNA-binding activity at

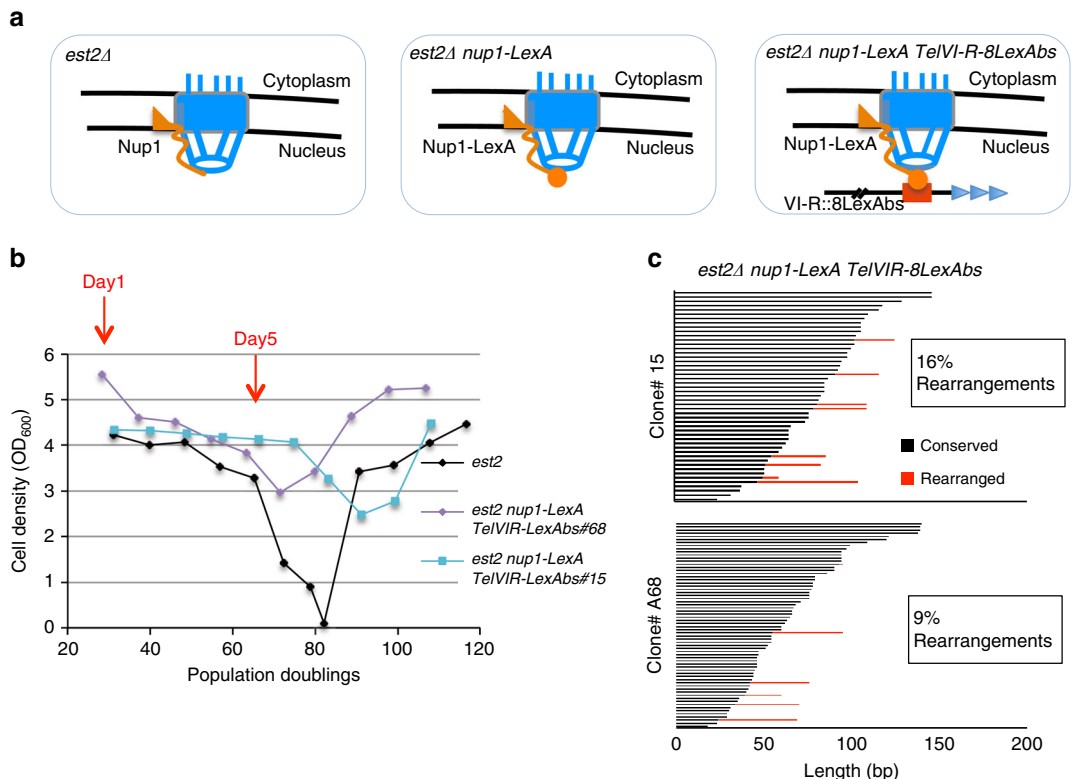

**Fig. 6 Tethering of *TelVI-R* to Nup1-LexA prevents SCR. a** Schematic of the three *est2Δ* strains expressing (right panels) or not (left panel) *nup1-LexA* and bearing either wild-type *TelVI-R* or modified *TELVI-R* with eight LexA DNA-binding sites inserted 1.2 kb away from the TG$_{1-3}$ repeats[35]. **b** Senescence profiles of the two *est2Δ nup1-LexA TelVI-R-8lexAbs* clones (#15 and #A68) used for sequencing. The *est2Δ* profile is from Supplementary Fig. 4 and is shown for comparison. D1 and D5 are highlighted in red. **c** *TelVI-R* sequence analysis of the clones *est2Δ nup1-LexA TelVI-R-8lexAbs* #15 and #A68 at Day 5 (see Supplementary Fig. 5 for the statistics of comparison with WT *TelVI-R*).

the NPC. We therefore made the assumption that non-specific DNA binding by LexA could interfere with telomere anchoring. We tested whether the overexpression of LexA from a multicopy vector could affect the continuous growth of *est2Δ nup1-LexA* cells. As shown in Supplementary Fig. 7e, overexpression of LexA did not affect the behavior of *nup1-LexA* cells in the absence of telomerase. This result prompt us to test whether the addition of the LexA DNA binding domain to Nup1 impairs a specific Nup1 function required to anchor stalled replication forks to the NPC.

We therefore tested whether loss of function mutants of Nup1 could phenocopy the *nup1-lexA* phenotypes. Expression of Nup1ΔFxFG in which most FG repeats have been removed[54] neither attenuated senescence (Fig. 7a) nor prevented survivor formation (Fig. 7b). In contrast, we found that expression of Nup1Δ1040–1076 (thereafter termed Nup1ΔCt) in which the last 36 residues at the C-terminal end have been removed[55], attenuated senescence in the absence of telomerase to an extent similar to that observed when Nup1-LexA is expressed (Fig. 7c). Accordingly, telomere length analysis showed that, as expected, telomeres shortened as a function of the PDs in the absence of telomerase but unlike *est2Δ* with wild-type *NUP1*, the *est2Δ nup1ΔCt* clones maintained short terminal TG$_{1-3}$ repeats much longer before displaying some Y′ amplification at late time points. The similarity with the phenotype of *nup1-LexA* prompted us to test whether Nup1ΔCt also affected the relocalization of replicative damages to the NPCs. Figure 7e, f shows that Nup1ΔCt impairs the peripheral enrichment of both the Cdc13-YFP/Rfa1-CFP foci and expanded CAG trinucleotide repeats in *est2Δ* cells.

Based on these results, we concluded that a loss of function of the Nup1ΔCt protein confers the same phenotypes as those observed in the *nup1-lexA* strain. We inferred that the LexA

fusion somehow impairs a Nup1 anchoring function associated with its C-terminal domain.

## Discussion

Studies over the last years revealed a prominent role of DNA damage relocalization to the nuclear pore in the repair of DNA lesions as diverse as irreparable DSBs, eroded telomeres, collapsed replication forks and replication forks stalled at DNA secondary structures[29,34,37,56]. Although several actors of this pathway have been identified, the functional consequences of the relocalization of damage to the NPC remained difficult to address due to the pleiotropic effects of inactivating pore components[42]. Here we show that disturbing the function of the basket protein Nup1 either by truncation of its C-terminal end or through a fusion with the bacterial DNA-binding protein LexA impairs the localization at the nuclear periphery of replication forks stalled at either expanded CAG tracts or telomeres.

We found that expression of *nup1-lexA* in cells deprived of telomerase activity profoundly altered the way their telomeres were processed. We demonstrated that interference with the Nup1 C-terminus unleashes a telomere maintenance mechanism that preserves a minimal telomere length compatible with proliferation. This iterative maintenance of TG$_{1-3}$ repeats by SCR prevents the production of very short and dysfunctional telomeres and hence permanent activation of the checkpoint and replicative senescence. It is not processive enough, however, to produce long telomeres as observed in type II survivors. Remarkably, Nup1-LexA expression in telomerase-negative cells renders cell growth insensitive to *RAD5* deletion, which normally accelerates senescence (Supplementary Fig. 4 and ref. [47]). Thus, the repair of telomeres that operates specifically in *est2Δ*

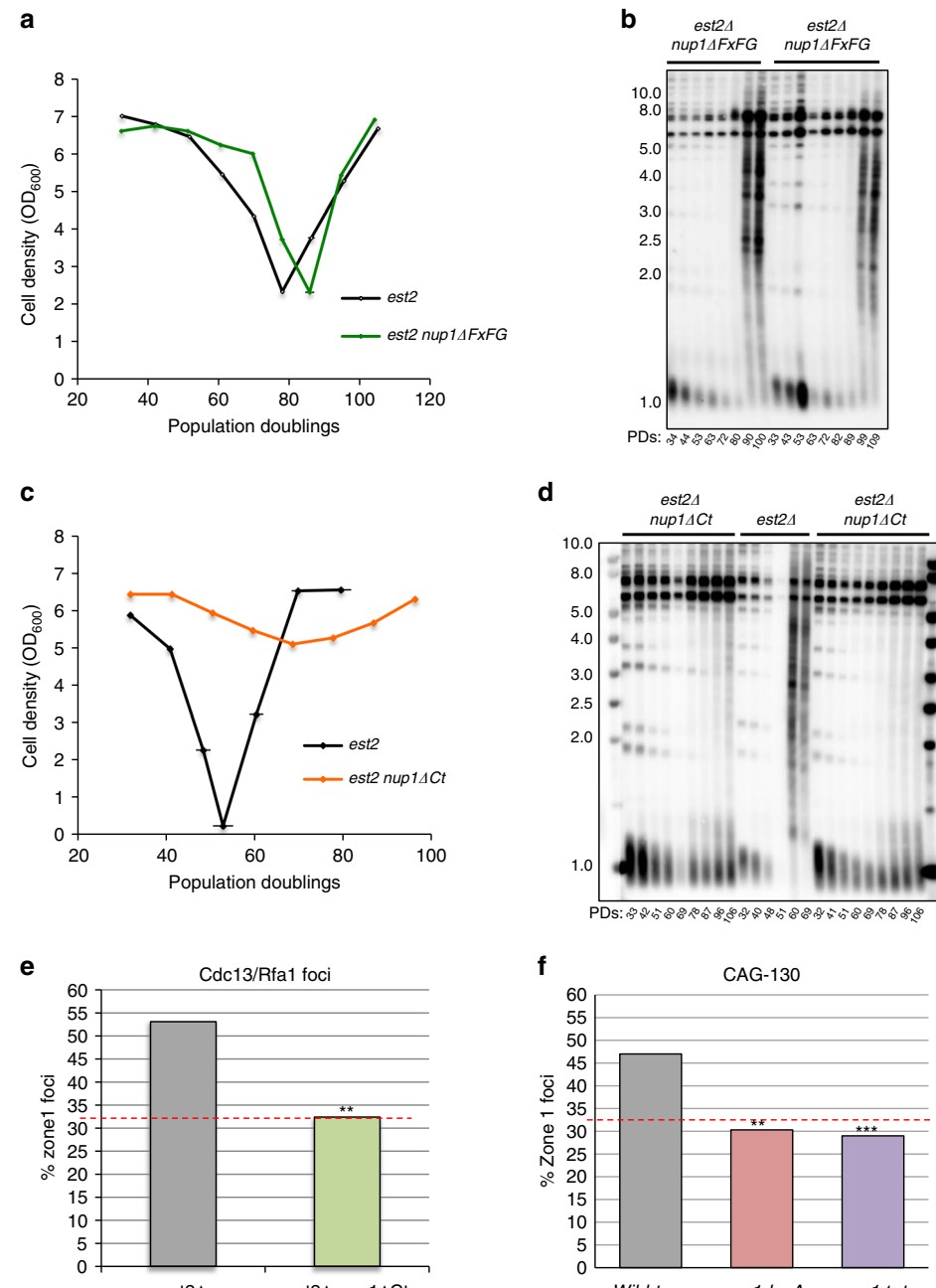

**Fig. 7 nup1ΔCt phenocopies the phenotypes of nup1-LexA. a** Mean senescence profiles of *est2Δ* (*n* = 5) and *est2Δ nup1ΔFxFG* (*n* = 5) clones. Error bars are SD. **b** Telomere length analysis of two representative *est2Δ nup1ΔFxFG* clones. Southern blot of *Xho*I-digested DNA prepared from samples of senescing cells was revealed with a TG$_{1-3}$ probe. **c** Mean senescence profiles of *est2Δ* (*n* = 4) and *est2Δ nup1ΔCt* (*n* = 8) clones. Error bars are SD. **d** Telomere length analysis of representative clones of the indicated genotypes. **e** The localization of the Cdc13-YFP/Rfa1-CFP foci in the three zones of equal area of the nucleus marked by Nic96-RFP was scored in late S and G2/M *est2Δ* and *est2Δ nup1ΔCt* cells as described in Fig. 2. The percentages of zone 1 foci are from three independently isolated *est2Δ nup1ΔCt* (*n* = 139 cells) clones. WT cells are from Fig. 3. Statistical differences were determined by a Fisher's exact test (**$p$ = 0.0019). **f** Percent of zone 1 foci for CAG-130 S-phase in *nup1ΔCt* cells (*n* = 143). Statistical differences were determined by a Fisher's exact test (***$p$ = 0,0006). WT and *nup1-LexA* cells are from Fig. 3. The repartition of the Rfa1/Cdc13 and CAG-130 foci between the three zones of the nucleus is shown in the Supplementary Fig. 8.

*nup1-LexA* cells seems to be independent of the error-free Rad5-dependent branch of the DDT pathway[57]. Instead, our data suggest that in this context, telomeres can be repaired by low-fidelity HR using the sister chromatid as a template. We emphasize that the HR pathway used in this process differs from the pathways leading to survivor formation. First it does not generate long telomeres as observed in type II survivors[25]. Second, it is detected soon after inactivation of the telomerase in

contrast to type I and II recombination events that occur at short telomeres during crisis. Finally, the HR at play in *est2Δ nup1-LexA* cells uses sister chromatid as a template instead of another telomere.

We propose that during replicative senescence, NPC relocalization of the forks stalled at telomeres facilitates their repair by a conservative mechanism (Fig. 8, left panel). This pathway might involve the Rad5-dependent branch of the DDT pathway to

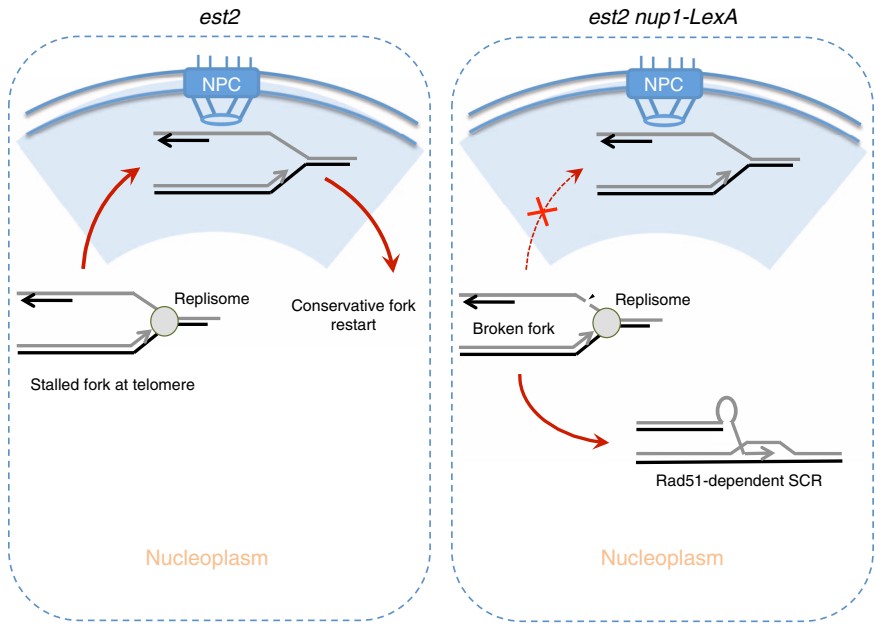

**Fig. 8 Model for the role of the NPC in the repair of fork stalled at telomeres.** Replication forks stalled at telomeres relocalize to the NPC and are mainly repaired by a conservative pathway that allows replication to resume (left). When this pathway fails, the stalled forks engage in error-prone Rad51-dependent SCR that maintain a telomere length compatible with continuous proliferation.

facilitate fork restart since the deletion of *RAD5* greatly accelerates senescence[47]. When relocalization to NPCs is impaired, an unrepaired stalled fork might be further processed and possibly break, allowing a default repair through SCR (Fig. 8, right panel). The structure of the fork that localizes to the NPC is unclear, raising the question of the difference between telomeric stalled forks and eroded telomeres, their recognition, and intranuclear partitioning.

Unexpectedly, preventing relocalization of damaged telomeres benefits the cells proliferating in the absence of telomerase, essentially abolishing replicative senescence. It may seem counterintuitive that cells developed a mechanism that actually increases the rate of senescence but yeast cells have evolved for best fitness in the presence of telomerase and do not experience any selective pressure for improved growth in the absence of telomerase. We showed that tethering of stalled fork to the pore favors conservative (error free) fork restart. Selection pressure most probably was on this aspect that is critical for genome stability in telomerase-positive cells. In this respect, the effect of the Nup1-LexA fusion is not limited to influencing the mechanism of telomere maintenance in telomerase-negative cells since it also increases the instability of CAG-triplet repeats (Fig. 3c). It is well established that long CAG tracts stall replication forks due to formation of hairpin structures[50,58,59]. Expanded CAG tracts transiently associate with the nuclear pore during late S phase in budding yeast, which prevents HR-dependent instability and may promote replication fork restart[37]. Previous work indicates that inactivation of the Nup84 protein prevents relocalization of triplet repeats to the NPCs and increases CAG repeat instability, especially Rad52-dependent contractions. Strikingly, we show here that Rad52-dependent CAG repeat contractions are also increased in the *nup1-LexA* strain. Therefore, the role of the NPC in restricting error-prone recombination between repeated sequences appears to be acting at two different types of fork-stalling sequences, telomeres and CAG/CTG repeats. This mechanism may be a universal way to control recombination during repair of stalled or collapsed forks to protect genome integrity.

The question remained why fusion of LexA to Nup1 affects the relocalization of the stalled forks to the NPCs? We discovered that truncation of the 36 C-terminal residues of Nup1 impacts the senescence profile and the localization of stalled forks at telomeres and at CAG triplet repeats at a similar extent as the C-terminal LexA fusion. Remarkably, the function of Nup1 in the processing of stalled replication forks appears to be specific since inactivation of the other basket proteins, i.e., Nup60 or Mlp1 and Mlp2, does not attenuate telomere-induced senescence (Supplementary Fig. 9a and ref. [35]). It is possible that the C-terminal domain of Nup1 may interact directly with one of the proteins bound to the stalled forks. Currently, the only function attributed to the unstructured C-terminal domain of Nup1 is that it contributes to the high-affinity interaction of Nup1 with the Kap60-Kap95-cargo complexes, although the way it does so remains elusive[55,60]. It has been proposed that the strong interaction of Nup1 with Kap95-Kap60 might help to guide the cargo to the correct site[60]. Although the *nup1-LexA* and *nup1ΔCt* alleles differ by their genetic interaction with the deletion of *nup60Δ* (Supplementary Fig. 9b, c), suggesting that interactions with the karyopherins is less affected by the LexA fusion than by the C-terminal deletion, one exciting possibility is that Nup1 regulates a secondary role of the karyopherins in escorting a cargo to the site of replication stress. Such docking functions, independent of the transport function per se, have already been attributed to karyopherins in tethering Ulp1 to the NPC[61] or in stabilizing the Dam1 complex until it associates with spindle microtubules[62]. In this scheme, Nup1-Kap-cargo interaction would contribute to the loading of the cargo onto the stalled replication fork either directly at the NPC or in the nucleoplasm before relocalization to the nuclear pore, whereby shaping the fork for conservative restart. Understanding the way Nup1-LexA and Nup1ΔCt interfere with the stalled forks attempting to interact with the NPCs will require further investigation.

Numerous actors participate in the response to DNA replication perturbations. Together, our data reinforce the pivotal role of the NPC to favor the most conservative pathway to rescue stalled replication forks. The need to preserve the fork against

recombination is probably more critical at repeated sequences where faulty annealing can lead to instability. It remains unclear whether the role of the NPC upon replication stress is preserved in metazoans but intranuclear positioning also participates in DNA repair pathway choice in human cells[63]. Repair pathway control might be fundamental for human health considering the role of trinucleotide repeat expansion in several diseases[64]. This is also true at telomeres since the iterative recombination between sister chromatids described here can thwart senescence, a potent anti-tumor mechanism in mammalian cells. Interestingly, a recent survey of 6835 cancers showed that 22% of tumors neither express telomerase at detectable level nor harbored characteristics of ALT[65]. In line with this unexpected observation, cases of metastatic melanoma cells[66] and high-risk neuroblastoma[67] have been reported that lacked significant canonical telomere lengthening mechanisms. We propose that unleashed SCR might constitute the first step in bypassing the proliferation barrier induced by unprotected telomeres during carcinogenesis, before full telomere stabilization takes place via either telomerase reactivation or ALT.

## Methods

**Yeast strains**. Strains used in this study are described in Supplementary Table 1. Strains were constructed and analyzed by standard genetic methods.

**Senescence assays**. Liquid senescence assays were performed starting with the haploid spore products of diploids that were heterozygous for *EST2* (*EST2/est2Δ*) and for the gene(s) of interest. To ensure homogeneous telomere length before sporulation, each diploid has been propagated for at least 50 PDs on YPD plates. After 2–3 days of growth at 30 °C, the entire spore colonies were transferred to 2 ml liquid YPD to estimate the number of PDs and the suspension immediately diluted to $10^5$ cells per milliliter. Cells were serially passaged in 15 ml of liquid YPD medium at $10^5$ cells per milliliter at 24-h intervals. Replicative senescence curves shown in this study correspond to the average of several independent spores with identical genotype. Senescence assays on solid medium were initiated as described above, but the cells were propagated by consecutive restreaking on solid YPD plates followed by outgrowth for 2 days at 30 °C. The process was repeated until the appearance of survivors.

**Telomere sequence analysis**. The telomere *TELVI-R* was amplified by PCR using the AccuPrime™ GC-Rich DNA Polymerase (InVitrogen) and the *TELVI-R* specific primer 5′-CGTGTGCGTACGCCATATCAATATG-3′ and cloned into TA-cloning vector (InVitrogen)[24]. Sequencing was perfomed by GATC BIOTECH. Telomere sequences were aligned using EMBOSS needle website. Conserved sequences were defined when perfectly matching to the consensus and upon single point mutation or 1–2-bp deletion/insertions. The point of divergence was defined when the sequence could not be aligned with the reference sequence.

**Telomere southern blot analysis**. A total of 25 µg of genomic DNA was digested with XhoI overnight at 37 °C. Digested DNA was resolved in 0.9% agarose gel and transferred onto an XL nylon membrane. The DNA was hybridized with a radiolabeled DNA fragment composed of $TG_{1-3}$ repeats.

**Fluorescence microscopy**. Fluorescence in situ hybridization (FISH) was performed as described[68]. Cells grown in YPD at 30 °C were fixed with 4% final paraformaldehyde to the media. The percentage of cells with poly(A)+-RNA accumulation was calculated from at least 250 cells per condition. Protein import and export from the nucleus were monitored using plasmids encoding GFP-NLS or GFP-NLS-NES[69], respectively. Observations of exponentially growing cells were performed using a Nikon Eclipse Ti microscope with a 100× objective. Cell images were captured with a Neo sCMOS Camera (Andor). For each field of view, 11 stacks were acquired at 0.3-µm intervals along the Z-axis. Because the GFP-NLS tends to form aggregates in the nucleus, total fluorescence intensity of each cell was measured as well as the fluorescence in the nucleus using ImageJ software. The nuclear/cytoplasmic ratio (N/C) was calculated on maximum intensity projection for each cell as ((nuclear fluorescence−background)/nuclear area) / ((total fluorescence−background) − (nuclear fluorescence−background) / (cell area−nuclear area)). For GFP-NLS-NES, the nucleocytoplasmic ratio was measured at the focal plane using the ROI plugin of ImageJ. Mean fluorescence intensity within constant square regions placed in the cytoplasm, the nucleus marked with NIC96-RFP and in the intercellular background was measured. N/C ratios are (Nucleus−background) / (Cytoplasm−background).

*Cdc13-YFP/Rfa1-CFP zoning assay*: Spore colonies from a heterozygous *est2Δ nup1-LexA CDC13-YFP RFA1-CFP NIC96-RFP* diploid were streaked once on YPD plates supplemented with adenine (6 mg per milliliter) and grown for 48 h (about 35 PDs from the original spore). Cells were then grown overnight at low density on YPD plates supplemented with adenine before imaging. Observations were performed using a Nikon Eclipse Ti microscope with a 100× objective and a binning 2×. Cell images were captured with a Neo sCMOS Camera (Andor) as Z-stack of 21 images with a step interval of 0.2 µm. RfA1-CFP and Cdc13-YFP co-localization was assessed at each Z-plane. The zoning assay was then performed using median filtered Rfa1-CFP and Nic96-RFP stacks at the Z-plane of Cdc13/Rfa1 co-localization using the point-picker plugin in ImageJ software[41]. Late S-phase cells were defined as cells with a medium bud and a single nucleus in the mother cells. G2/M cells were defined as cells with the nucleus at or spanning the bud neck. As the nuclei are often bi-lobed between the mother and the daughter at the G2/M transition, only the lobe containing the foci was considered for measurement.

*CAG repeat zoning assay*: Colonies were checked for presence of 130 CAG repeats by PCR with primers flanking the repeat. Cells from colonies with the correct repeat length were grown to ~$5 \times 10^6$ cells per milliliter in YC media, and a portion was re-checked to confirm the tract length had been maintained. Cells were fixed with 4% paraformaldehyde. Z-stack images were taken using a Zeiss AX10 fluorescent microscope under 100× magnification. Step interval size was 0.15 µm and ~30 images were taken per stack. Exposure time was DIC: 100 ms; GFP: 500 ms. Images were deconvolved, and three-zoning criteria was used to evaluate the location of the GFP foci for S-phase cells with the ImageJ point picker program[41]. S-phase cells were determined by yeast morphology as described in ref. 70. Data for each assay were obtained from two independent *nup1-LexA* strains, and three independent experiments were performed (two for strain #4470 and one for strain #4469).

**Triplet repeat stability and fragility assays**. The CAG tract was amplified from yeast colonies using primers spanning the repeats (P1 and P2 in Supplementary Fig. 3a) as described in Sundararajan et al.[71] to confirm correct tract length. Colonies were grown in YC-Leu liquid media for six to seven cell divisions to allow expansion, contraction or breakage, and were plated on FOA-Leu and YC-Leu plates. For the instability assay, CAG repeat length in at least 100 daughter colonies from at least four different parent colonies on the YC-Leu plates was assessed by PCR amplification and sizing of the amplicons on an AATI fragment analyzer system. Fragments 3 or more repeats longer or shorter than the starting tract length were counted as expansions or contractions, respectively. Data for each assay were obtained from two independent *nup1-LexA* strains. To assay fragility, colonies growing on FOA-Leu and YC-Leu were counted, and a rate of mutation was calculated using the method of the median[72]. At least three independent 10 colony fluctuation assays were performed per strain. A subset of FOA$^R$ colonies was checked for end loss (primers P3 and P4 in Supplementary Fig. 3a) and this occurred at a similar frequency in wild-type and *nup1-LexA* mutants.

**Reporting summary**. Further information on research design is available in the Nature Research Reporting Summary linked to this article.

## Data availability

The data that support the findings of the current study are available from the corresponding authors on reasonable request. The individual senescence curves underlying Figs. 1b–e, 7a, c and Supplementary Figs. 5b, 6a, b, 7a–c, e, and 8 as well as the uncropped and unprocessed scans of the Southern blots of the main figures are provided as a Source Data file.

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

## Acknowledgements

We thank Michael Lisby, Susan Wente and Hannah Klein for generous gifts of strains, Anna Babour for help with FISH experiment and Marion Dubarry, Daniel Isnardon and Davide Normanno for help with microscopy and processing of images. P.A. was supported by the Région Provence-Alpes-Côte d'Azur and the Association pour la Recherche contre le Cancer (ARC). C.M. was supported by L'Institut National Du Cancer (INCA, PLBIO14–012). V.G. is supported by the Ligue Nationale Contre le Cancer (équipe labellisée) and by INCA (PLBIO14–012). C.H.F. funding for this work was from NIH R01 grant GM122880.

## Author contributions

P.A., C.M., D.C. and M.N.S. designed and performed all experiments except the CAG trinucleotide repeat experiments that were designed and performed by J.W. and C.F. D.C. and C.F. discussed the results and contributed to the writing of the manuscript. M.N.S. and V.G. conceived this study and wrote the manuscript.

## Competing interests

The authors declare no competing interests.
