## [Peer Review File · Nature Communications]

Reviewers' comments:

Reviewer #1 (Remarks to the Author):

The work by P. Aguilera et al explores the consequences of expressing a modified form of NUP1, a major nucleoporin, on telomere damage repair, which is known to occur at the nuclear pore. They chose to fuse the bacterial LexA protein to the C-terminus of NUP1 at its own locus in a context where telomerase was abolished. Contrary to what is observed in a NUP1 wt context, telomerase negative cells carrying the NUP1-LexA fusion protein did not enter senescence but proliferated almost continuously in spite of rampant telomere shortening. Furthermore, telomeres were not recruited to the nuclear pore and the emergence of type I or type II survivals was either delayed or suppressed altogether. This surprising observation prompted the analysis of the mechanism that allowed telomere repair. First, the authors show that to escape from senescence NUP1-LexA cells require key homologous recombination factors. Second, the authors show that telomeres undergo a particular type of rearrangements resulting in repeat deletions or additions. Careful analyses revealed that these contracting or expanding events occurred likely through the imperfect annealing of an exposed overhang with the replicated double stranded portion of the sister telomere. Interestingly, similar contracting events also occurred when a different type of repeated and difficult to replicate sequence (long CAG/CTG tracts) was tested, which also failed to localize to the nuclear pore in the presence of NUP1-LexA. The authors further show that telomerase negative NUP1-LexA cells require intact RAD53 kinase activity. Oddly enough, only another DNA binding protein, and not non-DNA binding ones, fused to NUP1 was able to reproduce the effects observed with NUP1-LexA.

In all this is a very interesting work revealing a previously undescribed mechanism of telomere maintenance. The evidences are compelling and the conclusions are warranted by the results.

Comments:

- 1) There is no detailed description of the construction of the fused NUP1-LexA in this manuscript, so it is not known precisely how many amino acids were added to the NUP1 protein. However, knowing that the LexA protein is 202aa, the scheme representing the fused protein in figure 1a does not reflect this length. A more realistic scheme should be presented, indicating the total length of the fused protein.
- 2) Can the authors use a version of LexA or LacI unable to bind DNA (because of point mutations) to check whether DNA binding is really required?
- 3) The authors found that RAD53 dysfunction prevents senescence bypass by telomerase negative NUP1-LexA cells. Are telomeres addressed to the nuclear pore in this context?
- 3) The authors envision (Fig. 5c) that the products of sister telomere recombination in the NUP1-LexA cells may result in elongation (through motif duplication) as well as in shortening (through motif deletion). However, it is not clear what prevents telomeres from acquiring extended lengths. Can the authors discuss this?
- 4) In the same figure, the authors list factors required for break induced replication, thus suggesting that a similar mechanism is at work here. Is Pif1 required?

Reviewer #2 (Remarks to the Author):

The manuscript in question presents a very interesting observation. Specifically, the research team identified that tagging the *S. cerevisiae* nuclear pore basket protein Nup1 with a LexA DNA binding domain influences telomere and triplet expansion repeats. This is somewhat unexpected, but the presented data strongly indicate that altering Nup1 in this way prevents telomere and triplet repeat location to the nuclear periphery, impacts the length/stability of these repeats, alters recombination at these repeat sequences, and impacts telomere-dependent proliferation limits. The implication of these data is that preventing telomere relocation to the nuclear periphery engages alternative recombination mechanisms that facilitates telomere length homeostasis. While the presented data are intriguing, the paper would benefit from a better understanding and

description of the underlying mechanism. There are also points of confusion this reviewer needs clarified before publication. Specific comments are presented below.

1. The most substantial weakness in this paper is a lack of understanding as to why NUP1-LexA impacts these outcomes on telomeres and triplet repeats. It stands to reason it is because LexA prevents association of some telomere or repair factor with the periphery. What factors are involved? How is this mediated? Why does LexA do this when attached to NUP1? There needs to be a mechanistic link established here to clarify how LexA addition to NUP1 mediates changes in recombination pathways. Without this link, the data are interesting but incomplete.

2. In the manuscript the authors describe outcomes at two different telomere states. 1: Eroded telomeres following growth of over 60 PDs without telomerase (e.g. growth curves and telomere length measurements throughout) and 2: stochastic replication problems at normal length telomeres (imaging in figure 2, sequencing figure 5). NUP1-LexA prevents senescence caused by telomere shortening (state 1). However, the localization of telomeres to nuclear periphery was done on normal length telomeres with and without expanded CAG repeats and attributed to replication problems (state 2). Can the authors differentiate between NUP1-LexA impact on telomere outcomes at short telomeres vs. replication stressed telomeres, or are these two telomere states one in the same?

3. In figure 2, It is unclear to me why a single cdc13 and Rfa-1 focus is observed? Is this because all or a subset of the 32 telomeres in the yeast cell are co-localizing at a single point? Or is it because only one telomere is both Cdc13 and Rfa-1 positive? For figure 3 I understand as you are tethering the fluorescent protein to a single locus. However, I struggle to conceptualize the quantitative data in figure 2 without knowing what is being visualized with fluorescent microscopy.

4. The authors state that they have revealed a new mode of telomere maintenance (Page 11, line 11). How can they distinguish that NUP1-LexA expression induces recombination different than observed with Type 1 survivors? Clones #A11 and #A45 in Figure 4B show prominent Y' amplification. To me this appears to suggest preventing telomere association with the periphery drives a known outcome of recombination that leads to telomere maintenance and Type 1 survivors. It would be helpful to better characterize the pathway described in this manuscript in comparison to Type 1 or 2 survivors. For example, is the pathway described here activated together with the canonical recombination-based telomere length maintenance in survivors, or is it a completely novel mode of recombination? The authors should sequence individual telomeres from type-1 and 2 survivors to determine if similar sister telomeres recombination is observed.

5. Figure 5B - The description of how sequences are aligned and the data indicating how this verifies sister-telomere vs. non-sister telomere recombination needs to be better explained. I had difficulty understanding the logic of how their observations ruled out non-sister telomere interactions. Also, I did not understand how deletions occurred within a specific motif, or the sequence of that motif.

6. The est2delta NUP1-LexA TelVIR-8LexAbs data are inconclusive. The outcomes are of varying significance (one clone significant, one not-significant). Do these data have sufficient power to determine the effect of tethering a telomere to the pore?

Reviewer #3 (Remarks to the Author):

This group and others have shown that eroded telomeres naturally relocate at the nuclear periphery and attach to NPC. In this study Aguilera et al., show what happens to telomeres in the absence of telomerase if they do not relocalize to the pores.

They use a fusion of the nucleopore protein NUP1 with the DNA binding factor LexA that although

it does not globally affect the function of the pore it does prevent the eroded telomere from re localizing to the pore.

They find that shortened telomeres surprisingly do not elicit senescence and that they maintain their length by inter sister recombination.

They propose a model in which eroded telomere NPC localization promotes senescence and as a resort mechanism BIR dependent typeII survivors.

Although there are some interesting aspects in the study, yet there are issues that need to be resolved.

1. The study is based on the observation that fusing a DNA binding protein on NUP1 affects the relocation of shortened telomeres. The authors admit that they do not know the mechanism by which this happens. It is therefore strange to base an entirely study on a system not well understood. Can they prove their observations by another way?

2. Is the DNA binding activity of LexA important? Are the results the same in the presence of LexA mutated at the DNA binding domain?

3. Can the authors explain how they assess telomere location relative to the periphery? Yeast telomeres are already at the periphery.

4. How does this strain compare to Siz1/2 double mutant where the re localization to the pores is also inhibited?

5. What happens to the SUMOylation at the eroded telomeres? Is it increased?

We thank the referees for their constructive comments. You will find here below our point-by-point answer to the referee's concerns.

We want to stress out that the main novelty of the revised manuscript (in response to a concern raised by the 3 referees) is the discovery that deleting the last 36 amino-acids of Nup1 (the C-terminal region after the FG-rich domain) produces a mutant (*nup1 Δ Ct*) that recapitulates the phenotypes of the *nup1-lexA* mutant. In particular the *nup1- Δ Ct* mutant does not senesce, maintains its telomere to a short but stable length, and does not relocate neither the expanded CAG repeats nor the telomeric stalled forks to the NPC exactly as did the *nup1-lexA* mutant. These new findings indicate that the phenotypes that we observed with the *nup1-lexA* mutant were due to a loss a function of Nup1 as we suggested in the discussion of the MS. The details are in our point-by-point response.

Reviewer #1 (Remarks to the Author):

The work by P. Aguilera et al explores the consequences of expressing a modified form of NUP1, a major nucleoporin, on telomere damage repair, which is known to occur at the nuclear pore. They chose to fuse the bacterial LexA protein to the C-terminus of NUP1 at its own locus in a context where telomerase was abolished. Contrary to what is observed in a NUP1 wt context, telomerase negative cells carrying the NUP1-LexA fusion protein did not entered senescence but proliferated almost continuously in spite of rampant telomere shortening. Furthermore, telomeres where not recruited to the nuclear pore and the emergence of type I or type II survivals was either delayed or suppressed altogether. This surprising observation prompted the analysis of the mechanism that allowed telomere repair. First, the authors show that to escape from senescence NUP1-LexA cells require key homologous recombination factors. Second, the authors show that telomeres undergo a particular type of rearrangements resulting in repeat deletions or additions. Careful analyses revealed that these contracting or expanding events occurred likely through the imperfect annealing of an exposed overhang with the replicated double stranded portion of the sister telomere. Interesting, similar contracting events also occurred when a different type of repeated and difficult to replicate sequence (long CAG/CTG tracts) was tested, which also failed to localize to the nuclear pore in the presence of NUP1-LexA. The authors further show that telomerase negative NUP1-LexA cells require intact RAD53 kinase activity. Oddly enough, only another DNA binding protein, and not non-DNA binding ones, fused to NUP1 was able to reproduce the effects observed with NUP1-Lex A. In all this is a very interesting work revealing a previously undescribed mechanism of telomere maintenance. The evidences are compelling and the conclusions are warranted by the results.

Comments:

1) There is no detailed description of the construction of the fused NUP1-LexA in this manuscript, so it is not known precisely how many amino acids were added to the NUP1 protein. However, knowing that the LexA protein is 202aa, the scheme representing the fused protein in figure 1a does not reflect this length. A more realistic scheme should be presented, indicating the total length of the fused protein.

The scheme has been redrawn to scale accordingly (see the new Fig. 1).

2) Can the authors use a version of LexA or LacI unable to bind DNA (because of point mutations) to check whether DNA binding is really required?

To satisfy this concern, we have substituted alanine 42 and 43 with proline in the second α -helix of the LexA DNA binding domain. These two point mutations are predicted to decrease the DNA binding activity of LexA. The mutated LexA (called LexA_{mut}) was fused at the C-terminal end of Nup1. We found that the double substitution only partially reversed the attenuated replicative senescence (see the enclosed figure at the end of the point by point response) indicating that the DNA binding activity contributes to the phenotypes conferred by the Nup1-LexA. However, this result has not been added in the revised version of the manuscript. Instead, we present new results showing that a Nup1 mutant lacking the last 36 amino-acids (Nup1 Δ Ct) recapitulates the phenotype of nup1-LexA. These new results reveal that the common phenotypes associated with the expression of Nup1 Δ Ct or Nup1-LexA are likely due to a loss of function of Nup1. We think that having a DNA binding domain fused to Nup1 somehow induces this loss of function. We discuss the way by which the extreme C-terminus of Nup1 may affect the relocation of forks stalled at telomeres or at CAG expanded repeats to NPCs

3) The authors found that RAD53 dysfunction prevents senescence bypass by telomerase negative NUP1-LexA cells. Are telomeres addressed to the nuclear pore in this context?

We have results indicating that Rad53 is required for localization to NPCs of the extended CAG repeats, which are no longer detected at the NPCs in the absence of Rad53 (JW and CF unpublished results). It is therefore very unlikely that inactivation of Rad53 would restore localization of telomeres to NPCs in the nup1-LexA cells.

3) The authors envision (Fig. 5c) that the products of sister telomere recombination in the NUP1-LexA cells may result in elongation (through motif duplication) as well as in shortening (through motif deletion). However, it is not clear what prevents telomeres from acquiring extended lengths. Can the authors discuss this?

We think that the mechanism is not processive enough to produce very long telomeres as observed in type II survivors. Actually we don't think that iterative unequal exchange between sister chromatid is at the basis of the long telomere repeats in type II since type II survivors actually emerge when the telomere length is the shortest (at crisis, <70bp/telomere). We envision an other mechanism for type II recombination and have data suggesting that the generation of at least the first elongated telomeres do not use terminal telomeric repeats as template for telomere elongation. These issues are discussed in the revised version (p16).

4) In the same figure, the authors list factors required for break induced replication, thus suggesting that a similar mechanism is at work here. Is Pif1 required?

We performed this experiment and found that Pif1 is required. Indeed, the deletion of *PIF1* did reverse the senescence phenotype of the nup1-LexA mutant suggesting that the mechanism of BIR is involved in the new repair pathway that we describe here. This data has been added in the new version of the manuscript (suppl. Fig 4).

Reviewer #2 (Remarks to the Author):

The manuscript in question presents a very interesting observation. Specifically, the research team identified that tagging the *S. cerevisiae* nuclear pore basket protein Nup1 with a LexA DNA binding domain influences telomere and triplet expansion repeats. This is somewhat unexpected, but the presented data strongly indicate that altering Nup1 in this way prevents telomere and triplet repeat location to the nuclear periphery, impacts the length/stability of these repeats, alters recombination at these repeat sequences, and impacts telomere-dependent proliferation limits. The implication of these data is that preventing telomere relocation to the nuclear periphery engages alternative recombination mechanisms that facilitates telomere length homeostasis. While the presented data are intriguing, the paper would benefit from a better understanding and description of the underlying mechanism. There are also points of confusion this reviewer needs clarified before publication. Specific comments are presented below.

1. The most substantial weakness in this paper is a lack of understanding as to why NUP1-LexA impacts these outcomes on telomeres and triplet repeats. It stands to reason it is because LexA prevents association of some telomere or repair factor with the periphery. What factors are involved? How is this mediated? Why does LexA do this when attached to NUP1? There needs to be a mechanistic link established here to clarify how LexA addition to NUP1 mediates changes in recombination pathways. Without this link, the data are interesting but incomplete.

Initially, we were thinking that the LexA DNA binding domain could prevent the relocation of telomeric stalled forks by binding non-specifically with genomic DNA. We have performed several experiments to prove this model but none of these experiments were conclusive. Thus to further understand how nup1-LexA could affect localization to NPCs, we decided to test loss of function mutants of Nup1. Strikingly, we found that a truncation of the 36 C-term residues of Nup1 (Nup1 Δ Ct) recapitulates the most relevant phenotypes of the Nup1-LexA mutant ie a defect in the relocalization of both extended CAG repeats and damaged telomeres at the NPCs and an attenuated senescence (these results are shown in the new Fig. 7). The effect of the Nup1 Δ Ct was specific since expressing a Nup1 mutant in which most FG repeats have been removed neither attenuated senescence nor prevented survivor formation.

In the revised version, we now favor a model in which the C-terminal domain of Nup1 may interact directly or indirectly with one of the proteins bound to the stalled forks. We discuss what could be the function of Nup1 C-terminal domain knowing that the only function attributed to this unstructured region is that it contributes to the high affinity interaction of Nup1 with the Kap60-Kap95-cargo complexes, although the way it does so remains elusive.

We believe that the identification of this mutant will be of great value in the field. Indeed, in contrast to full deletion of nucleoporins that have been used up to now (nup84 Δ for example), the Nup1 Δ Ct (and the nup1-LexA) affect the relocation of stalled forks at telomeres or at expanded CAG without impairing the localization at the NPC of the SUMO-protease Ulp1.

2. In the manuscript the authors describe outcomes at two different telomere states. 1: Eroded telomeres following growth of over 60 PDs without telomerase (e.g. growth curves and telomere length measurements throughout) and 2: stochastic replication problems at normal length telomeres (imaging in figure 2, sequencing figure 5). NUP1-LexA prevents senescence

caused by telomere shortening (state 1). However, the localization of telomeres to nuclear periphery was done on normal length telomeres with and without expanded CAG repeats and attributed to replication problems (state 2). Can the authors differentiate between NUP1-LexA impact on telomere outcomes at short telomeres vs. replication stressed telomeres, or are these two telomere states one in the same?

According to our current model (supported by the data), telomeres that experience replication stress *regardless of their length* relocalize to the NPCs. Localization at NPCs is evident pretty much immediately after telomerase inactivation when telomeres are still substantially long. Importantly, this promotes restart of the stalled replication forks but *does not* prevent gradual telomere shortening in the absence of telomerase which drives yeast with wild-type Nup1 into senescence (see Xu, Z. *et al.* Two routes to senescence revealed by real-time analysis of telomerase-negative single lineages. *Nat Commun* **6**, 7680; 2015). We reported that when telomeres are highly eroded and resected, they are relocalized to the NPC by a SUMO-dependent mechanism (Churikov *et al.*, Cell Report, 2016).

When localization of telomeric stalled forks at NPCs is impaired in yeast expressing nup1-LexA, the replication forks stalled at telomeres engages in low-fidelity break-induced replication that uses sister chromatid as a template. This helps to maintain telomeres at the minimum functional length in telomerase-negative yeast. In such a case we do not obtain critically short telomeres. Therefore it is difficult to differentiate between nup1-LexA impact on telomere outcomes at short telomeres vs. replication stressed telomeres. We modified the text accordingly to better explain this point (pages 9-10).

3. In figure 2, It is unclear to me why a single cdc13 and Rfa-1 focus is observed? Is this because all or a subset of the 32 telomeres in the yeast cell are co-localizing at a single point? Or is it because only one telomere is both Cdc13 and Rfa-1 positive? For figure 3 I understand as you are tethering the fluorescent protein to a single locus. However, I struggle to conceptualize the quantitative data in figure 2 without knowing what is being visualized with fluorescent microscopy.

We have previously shown that one single telomere can produce a Cdc13-YFP focus in telomerase negative cells (Khadaroo *et al.* 2009). Appearance of Cdc13-YFP foci correlates with accumulation of single strand DNA. In contrast to *est2Δ* cells, *yku70Δ* cells that have extended G-tails show several Cdc13-YFP foci suggesting that the clustering of damaged telomeres is limited, if any. The text has been reworded to clarify this point.

4. The authors state that they have revealed a new mode of telomere maintenance (Page 11, line 11). How can they distinguish that NUP1-LexA expression induces recombination different than observed with Type 1 survivors? Clones #A11 and #A45 in Figure 4B show prominent Y' amplification. To me this appears to suggest preventing telomere association with the periphery drives a known outcome of recombination that leads to telomere maintenance and Type 1 survivors.

Previously, we showed that the first step of type I survivor formation is the acquisition of a Y' element by the "X-only" telomere (Churikov *et al.* 2014). Fig4B shows that X-only telomeres are maintained for the time of the experience in all three clones 6, A11 and A45. There is indeed appearance of Y' amplification at the late time points in clone A45 as mentioned in the text but not in the clone A11 (please note the lower loading of the 2 first time points in this

clone which is indeed misleading concerning the level of the Y' signals). We further emphasize that we sequenced a single X-only telomere (TEL-6R) using a specific primer localized just upstream the TG1-3 repeat. It is thus technically impossible that we sequenced a product of type I recombination in our experimental conditions. The mode of telomere maintenance in *nup1-LexA* cells is clearly different from the recombination pathway that leads to Type 1 survivors.

It would be helpful to better characterize the pathway described in this manuscript in comparison to Type 1 or 2 survivors. For example, is the pathway described here activated together with the canonical recombination-based telomere length maintenance in survivors, or is it a completely novel mode of recombination? The authors should sequence individual telomeres from type-1 and 2 survivors to determine if similar sister telomeres recombination is observed.

As discussed in the manuscript (page 16), the pathway we are describing here differs from type II recombination that is dependent on Rad59 and not Rad51 and produces very long telomeres. In addition it is difficult to imagine that very long Type II telomeres (up to 10Kb) arise through sister chromatid exchange at a time when telomeres are very short (about 50-70bp at crisis). This does not rule out the possibility that a mechanism similar to the one revealed here is not at play in already formed type II survivors. They would be however very hardly detectable on long telomeres only by sequencing.

5. Figure 5B - The description of how sequences are aligned and the data indicating how this verifies sister-telomere vs. non-sister telomere recombination needs to be better explained. I had difficulty understanding the logic of how their observations ruled out non-sister telomere interactions. Also, I did not understand how deletions occurred within a specific motif, or the sequence of that motif.

This point is based on a specific characteristic of *S. cerevisiae* where the telomere sequences are degenerate because the telomerase has the peculiarity to be poorly processive and adds TG1-3 repeats of variable length. As a consequence each telomere sequence is unique and stable over generations except for its more distal region if elongated by the telomerase or modified by recombination. This peculiarity has been used to analyze the mechanism of elongation by the telomerase (Teixeira et al., Cell 2004, Hardy et al., Nat com 2014, Clausin and Chang PLOS genet. 2016...). A telomere sequence coming from a non-sister telomere would not align at any position of TEL6R without mismatches. The text has been modified to better explain this point and avoid confusion (p10).

6. The *est2delta* NUP1-LexA TelVIR-8LexAbs data are inconclusive. The outcomes are of varying significance (one clone significant, one not-significant). Do these data have sufficient power to determine the effect of tethering a telomere to the pore?

We would like to emphasize that the difference between *est2Δ* NUP1-LexA and *est2Δ* NUP1-LexA TelVIR-8LexAbs is highly significant for the two clones analyzed (ie $p < 0,005$). In contrast, neither of the two *est2Δ* *nup11-LexA* TelVIR-8LexAbs is significantly different from the WT clones. Our conclusion was further reinforced by the observation that as expected tethering of TelVIR-8LexAbs to the pore in *nup1-LexA* cells also improved type II recombination as expected for telomeres getting very short. Since we are aware that the different recombination modes operating at telomeres during senescence can easily be confused, these data were not included in the first version of the manuscript to avoid

confusion between the SCR described here and type II recombination. They have now been included in the supplementary Fig.5 (panels c,d,e).

Reviewer #3 (Remarks to the Author):

This group and others have shown that eroded telomeres naturally relocate at the nuclear periphery and attach to NPC. In this study Aguilera et al., show what happens to telomeres in the absence of telomerase if they do not relocalize to the pores. They use a fusion of the nucleopore protein NUP1 with the DNA binding factor LexA that although it does not globally affect the function of the pore it does prevent the eroded telomere from re localizing to the pore. They find that shortened telomeres surprisingly do not elicit senescence and that they maintain their length by inter sister recombination. They propose a model in which eroded telomere NPC localization promotes senescence and as a resort mechanism BIR dependent typeII survivors.

Although there are some interesting aspects in the study, yet there are issues that need to be resolved.

1. The study is based on the observation that fusing a DNA binding protein on NUP1 affects the relocation of shortened telomeres. The authors admit that they do not know the mechanism by which this happens. It is therefore strange to base an entirely study on a system not well understood. Can they prove their observations by another way?

We totally agree with this remark. To answer the question “*can we prove our observations by another way?*”, we analyze the phenotypes of C-terminal mutants of Nup1 with the idea that the LexA moiety of the fusion protein could cause a loss of function of Nup1. As mentioned above to our response to Referees 1 and 2, we now show that a small C-terminal truncation of Nup1 recapitulates the most relevant phenotypes of the nup1-LexA mutant. Importantly, the phenotypes conferred by the Nup1 Δ Ct are specific since they are not observed in a Nup1 mutant in which most FG repeats have been removed. These results have been added to the revised manuscript (new Fig. 7) as well as the discussion of mechanisms by which the nup1-lexA and the Nup1 Δ ct could operate (please see our detailed answer to the point 1 of the Reviewer #2).

,

2. Is the DNA binding activity of LexA important? Are the results the same in the presence of LexA mutated at the DNA binding domain?

We performed the experiments to answer these points. Please see our answer to the point 2 of the Reviewer #1.

3. Can the authors explain how they assess telomere location relative to the periphery? Yeast telomeres are already at the periphery.

Yeast telomeres are indeed anchored to the nuclear membrane. We have previously shown that dysfunctional telomeres (marked by Cdc13 and Rad52 foci) are relocalized from their normal nuclear membrane anchor sites to the NPC (Khadaroo et al, 2009).

In this study, we aimed to assess the location of damaged telomeres only. Telomeric stalled forks accumulate single strand DNA that is covered by Cdc13 at the G-strand and RPA at the C-strand. We thus use the colocalization of Cdc13-YFP and Rfa1-CFP as a read-out of Telomeric stalled forks. By monitoring the co-localization Cdc13-YFP and Rfa1-CFP relative to the nuclear periphery (marked by Nic96-RFP), we observed dysfunctional telomeres are no longer anchored to the nuclear periphery in *est2Δ nup1-LexA* cells. Since we previously reported that dysfunctional telomeres were associated to the NPC, we infer that localization to NPC of those telomeres is lost.

Importantly, the localization of functional telomeres (marked by Rap1-CFP foci) to the nuclear periphery is not impaired in *nup1-LexA* cells (not shown).

4. How does this strain compare to Siz1/2 double mutant where the re localization to the pores is also inhibited?

We performed the experiments to answer this question and found that deletion of both *SIZ1* and *SIZ2* does not reverse the phenotype of the *NUP1-LexA* mutant during senescence. This suggests that Siz1- and Siz2-dependent SUMOylation is not required for the new mechanism of telomere maintenance that we have described.

Note also that unlike the *nup1-LexA* or the *Nup1ΔCt* mutants, the *siz1Δ siz2Δ* double mutant (that affects relocalization of highly eroded telomere to the NPCs) is unable to maintain telomeres at the minimal functional length but rather undergoes accelerated senescence in the absence of telomerase (Churikov et al., 2016). We think that that the *Nup1* defect, but not the absence of *Siz1* and *Siz2*, shape the fork in a way permissive for sister chromatid recombination (see discussion).

5. What happens to the SUMOylation at the eroded telomeres? Is it increased?

We have previously show that SUMOylation is increased at the eroded telomeres by the time of crisis. We have previously linked it to the process of eroded telomere relocalization to NPC (Churikov et al., 2016). Further study will be needed to determine whether SUMOylation is required to relocalize telomeric stalled forks to the NPCs.

(a) Mean senescence profile of *est2Δ NUP1-LexAmut* clones (n=12) compared with *est2Δ NUP1-LexA* clones (n=9). Error bars are SD. **(c)** Average cell density at the peak of senescence (OD₆₀₀) of *est2Δ NUP1-LexA*, *est2Δ NUP1-LexAmut* and *est2Δ* clones. P-values are Fisher's least significant difference (LSD) test. Error bars are SD. **(d)** Telomere length was analyzed by TG₁₋₃ probed Southern blot of *XhoI*-digested DNA prepared from samples undergoing replicative senescence. Two representative clones are shown. **(e)** Relative frequencies of survivor profiles in the different clones. « no survivor like », « Type I », « Type I/II », « Type II » refer respectively to the terminal recombination patterns of clones 6, A45, A11 and *est2Δ* shown in Fig. 2b.

REVIEWERS' COMMENTS:

Reviewer #1 (Remarks to the Author):

All my concerns were adequately dealt with by the authors. In addition, the new data showing that a C-terminal deletion of NUP1 is sufficient to recapitulate the phenotypes of a NUP1-LexA fusion convincingly demonstrate that this is an intrinsic property of NUP1 and not a new function brought in by a foreign protein.

Reviewer #2 (Remarks to the Author):

Thank you to the authors for clearly and concisely addressing the reviewers' comments. The added nup1-deltaCt data are a very nice addition to the paper. I recommend accepting the revised manuscript for publication.

Reviewer #3 (Remarks to the Author):

The authors have made a substantial effort to answer the reviewers' comments and I therefore recommend publication.